# Mediterranean Diet, a Posteriori Dietary Patterns, Time-Related Meal Patterns and Adiposity: Results from a Cross-Sectional Study in University Students

**DOI:** 10.3390/diseases10030064

**Published:** 2022-09-11

**Authors:** Paraskevi Detopoulou, Vassilis Dedes, Dimitra Syka, Konstantinos Tzirogiannis, Georgios I. Panoutsopoulos

**Affiliations:** 1Department of Nutritional Sciences and Dietetics, Faculty of Health Sciences, University of Peloponnese, New Building, 24100 Kalamata, Greece; 2Department of Clinical Nutrition, General Hospital Korgialenio Benakio, Athanassaki 2, 11526 Athens, Greece; 3Internal Medicine Department, Mediterraneo Hospital, 16675 Athens, Greece

**Keywords:** students, obesity, dietary patterns, meal patterns, breakfast

## Abstract

The transition to university is connected to potentially obesogenic dietary changes. Our aim was to assess the relation of Mediterranean diet adherence, and a posteriori dietary and meal patterns with adiposity in Greek students at the University of the Peloponnese. A total of 346 students (269 women) participated. Anthropometry was performed, and a food frequency questionnaire was administered. The MedDietScore was higher in women and was not linearly related to adiposity. Principal component analysis revealed six patterns: (1) legumes/vegetables/fruits/tea/dairy/whole grains, (2) juice/sodas/liquid calories, (3) olive oil/fats, (4) meat/poultry/fish, (5) alcohol/eggs/dairy and (6) fast foods/sweets. Patterns 4 and 6 were related to overweight/obesity probability (OR = 1.5, 95% CI: 0.995–2.538 and OR = 2.5, 95% CI: 1.07–6.06, respectively) and higher waist circumference (men). Men “early eaters” (breakfast/morning/afternoon snack) had a higher MedDietScore and lower overweight probability (OR = 0.47, 95% CI: 0.220–1.020). Poor meal and dietary patterns relate to overweight and central obesity, which is important for targeted health promotion programs.

## 1. Introduction

The transition to university is linked to increased autonomy, while different dietary habits and preferences may develop because of new exposures, socializing, food insecurity, lacking cooking skills, low self-efficacy and perceived stress [1,2,3,4,5]. At the same time, class schedules may render food preparation challenging and lead to skipping meals or consuming “convenient” unhealthy snacks [6,7]. Gender differences are also observed in several studies, with female students following diets rich in fruits and low in fat, eating breakfast at a higher frequency [8,9,10] and having more weight-loss attempts [11]. This unique blend of personal, socioeconomic and emotional factors makes this period critical for obesity and poor diet development [12].

Several studies have underlined the problems of increased adiposity and/or unhealthy dietary patterns in university students [5,13,14,15,16,17]. Indeed, the adoption of healthy dietary patterns, such as the Mediterranean diet [18,19,20], and other dietary features, such as meal frequency [21], main meal consumption and breakfast [6,22], have been connected to reduced obesity and increased muscle mass in the general population and young adults, although some studies have not shown any effect [23]. The diet–obesity hypothesis in this population becomes more complex when considering the role of underreporting, which has been addressed in only a few studies [24,25,26]. It is also noted that differences exist in the dietary habits of students living in different countries, as evidenced by multi-centered studies using the same methodology to ease between-country comparisons [18,23,27,28], suggesting that area-specific data are essential.

In Greece, data are scarce regarding the food habits of university students from Attika [5,29], and northern [3,30,31] and southern Greece [32,33]. The conducted studies mostly assessed food groups [1,5,18,32], nutrients [31,32] or a priori adherence to the Mediterranean diet [18]. Thus, a posteriori-defined dietary patterns as well as the effect of underreporting have not been addressed in Greek students and there is only one study from our group regarding the Peloponnese area [34].

Given the importance of the establishment of a healthy weight and healthy diet for cardiovascular disease prevention [35], the aim of the present work was to assess the relationship between (i) a priori dietary patterns (Mediterranean Diet Score), (ii) a posteriori dietary patterns and (iii) time-related meal patterns and obesity and central obesity indices in a generally young population of students in the University of the Peloponnese after controlling for underreporting.

## 2. Materials and Methods

### 2.1. Study Design

During the period 20 September 2018 to 10 November 2018 a cross-sectional survey was conducted on students of the University of the Peloponnese (departments of “Nursing”, “Sports Organization and Management”, “Philology” and “History, Archaeology and Cultural Resources Management”) or the former Technical Educational Institution of Peloponnese (field of Management, Agriculture and Food Technology). We followed a “convenience sampling” procedure, as described elsewhere [34].

Three hundred and forty six students participated in the study (269 women and 77 men), constituting 8.6% of the University students at the time of measurement. The age range was 18–47 years, and the students were enrolled in several semesters of their studies. The study protocol was approved by the University’s Ethics Committee (Faculty of Human Movement and Quality of Life Sciences). All procedures were in accordance with the Declaration of Helsinki (1989) of the World Medical Association, as revised in 2013. Informed consent was given by the participants and the president of each department.

### 2.2. Anthropometry

Weight and height were measured to the nearest 0.1 Kg and 0.1 cm, correspondingly [34], and the body mass index (BMI) was calculated as weight (in kilograms) divided by height squared (in meters squared). Individuals were then classified as underweight, normal weight, overweight or obese [36]. Waist circumference (WC) was measured with a tape to the nearest 0.1 cm between the superior iliac crest and the lower rib margin in the midaxillary line, after a moderate expiration. Hip circumference was measured to the nearest 0.1 cm as the maximal horizontal circumference at the level of the buttocks. Increased waist circumference was assessed according to International Diabetes Federation (IDF) criteria, WC > 94 cm for men and >80 cm for women [37].

### 2.3. Lifestyle Variables

Participants reported their weekly frequency of main meals and snack consumption. For each meal or snack (breakfast, morning snack, lunch, afternoon snack, dinner and bedtime snack) they checked the frequency of consumption: <once per week, 1–2 times per week, 3–4 times per week, 5–6 times per week or every day. Then, scores were attached to each frequency category from 1 to 5, correspondingly, and the total score was calculated (meal score).

### 2.4. Nutrition Assessment

A semi-quantitative food frequency questionnaire (FFQ) was developed for the purposes of the present study, as described elsewhere [34]. Briefly, it included 156 foods/beverages and 9 frequencies to choose, from “never” to “everyday” consumption. The portions used were 250 mL for milk, juices and other liquids, 30 g for cheese/ham, 200 g for yogurt, ½ cup for boiled vegetables, 1 cup for raw vegetables, ½ cup for rice and cereals, 1 cup for legumes, one medium fruit, 85 g for meat and 170 g for fish. The Cronbach’s alpha coefficient was 0.89 for internal consistency. Several food groups were considered, i.e., juices, sodas, liquid calories, alcoholic drinks, low-fat dairy (including cheese), high-fat dairy (including cheese), total dairy (sum of low-fat and full-fat dairy), vegetables, legumes, eggs, red meat, poultry, fish and seafood, refined grains, whole grains, honey and marmalade, fruits, sweets, other fats, olive oil, tea, coffee, nuts, fast foods. MedDietScore was used to assess Mediterranean diet adherence [38], with appropriate considerations in food portions if needed.

The USDA database was used to calculate energy intake [39]. Basal metabolic rate (BMR) was estimated using the Schofield equation for adults [40]. Underreporting was assessed by calculating the energy intake/BMR ratio and using the thresholds <1.09 for women and <1.07 for men, as previously suggested [41].

### 2.5. Statistical Analysis

Variables were tested for normality with the use of Kolmogorov–Smirnoff criterion. Normally distributed variables are presented as mean values ± standard deviation, while non-normally distributed variables as median and 25th–75th quartiles. For categorical variables absolute numbers and relative frequencies (%) are presented. For comparisons of normally distributed or transformed continuous variables between two groups, the *t*-test was applied. For comparisons of categorical variables across genders, the Chi-square test was used with Bonferroni corrections.

Two multivariate factor analyses using principal components analysis (PCA) were applied, for dietary patterns and meal patterns. To decide the number of components to retain from the factor analysis, the eigenvalues derived from the correlation matrix of the standardized variables were examined. Components with eigenvalue greater than 1 were kept for the data analyses. Moreover, the scree plot was considered. Six components of dietary patterns and 3 components of meal patterns were finally extracted. The Kaiser–Meyer–Olkin criterion and the Bartlett’s test were used to evaluate the factor’s analysis performance. Based on the principle that the component scores are interpreted similarly to correlation coefficients (i.e., higher absolute scores indicate that the food group contributes most to the construction of the component), the food patterns were defined in relation to scores of variables that correlated most with the component (absolute loading value > 0.45). For the identification of meal patterns, a total loading value > 0.60 was used to characterize a pattern. The orthogonal varimax rotation was used to derive optimal, non-correlated dietary patterns.

Linear regression models were developed to evaluate the association of the extracted food patterns or meal patterns (explanatory variables) with BMI and WC (dependent variables). BMI was logarithmically transformed and WC was transformed as 1/WC to achieve normality. Moreover, models were applied using BMI and WC in categorical forms (dichotomous variables) (BMI  ≥  25 kg/m^2^ (yes/no)), (high waist according to IDF criteria WC > 94 cm for men and >80 cm for women (yes/no)) [37]. Binary logistic regression models were performed to ascertain the effects of food patterns and meal patterns on the likelihood of overweight/obesity, by using a dichotomous dependent variable (BMI  ≥  25 kg/m^2^ (yes/no)). Analogous models were used to evaluate the effects of food patterns and meal patterns on the likelihood of increased WC, by using a dichotomous dependent variable according to the IDF criteria [37] (WC > 94 cm for men and >80 cm for women (yes/no)). The results were expressed as the exponential of the regression coefficient, which corresponds to an estimated odds ratio (OR). Adjustments were made for age, underreporting, department, living area (before enrollment) and gender (if applicable). All analyses were performed in the whole sample and in men and women separately, given the strong sex differentiation in students’ diets [8,9,10]. All reported *p*-values are based on two-sided tests and compared to a significance level of 5%. IBM SPSS Statistics for Windows version 22.0 (IBM Corp., Armonk, NY, USA) software was used for all the statistical calculations.

## 3. Results

### 3.1. Basic Characteristics of the Volunteers

The basic characteristics of the subjects are presented in Table 1. The greater participation rate was observed for the department of Nursing, followed by the departments of Philology, History, Archaeology and Cultural Resources Management and Sports Organization and Management. It is noted that 44.2% of the total sample were freshmen and 77.7% were women (*n* = 269 women). Women had a lower mean BMI and a lower rate of overweight than men. Moreover, women had a lower WC than men but scored higher regarding the sex-specific cutoff for high WC. It is noted that data on the year of study were missing for two students (one man and one woman) and that the living area was missing for 70 students (17 men and 53 women).

### 3.2. Food Group Consumption

The food intake of the volunteers is shown in Table 2. As can be seen, women had lower consumption of full-fat dairy, red meat, refined grains, eggs, nuts, fast foods, sodas and alcohol than men and higher vegetables, legumes and honey consumption than men. It is noted that men had a higher rate of underreporting than women.

### 3.3. MedDietScore

The median values and interquartile ranges (25th–75th percentile) of MedDietScore were 29 (24.5–32) for men and 30 (27.0–34.0) for women (*p* < 0.014), suggesting that women had a greater adherence to the Mediterranean diet compared to men. There was no relation between MedDietScore and BMI or waist circumference in multi-adjusted linear regression models or binary logistic regression models. It is noted that after ranking the variables, the MedDietScore was inversely related with BMI (partial rho = −0.148, *p* = 0.006) after adjustment for age, sex and underreporting, suggesting an inverse monotonic relationship.

### 3.4. Dietary Patterns

From the PCA analysis, six patterns were identified, i.e., pattern 1 (legumes, vegetables, fruits, tea, dairy, whole grain), pattern 2 (juice, sodas, liquid calories), pattern 3 (olive oil, fats), pattern 4 (meat, poultry, fish), pattern 5 (alcohol, eggs, dairy) and pattern 6 (fast foods, sweets). The observed patterns explained 60% of the total variance in the food groups’ intake (Appendix A).

#### 3.4.1. Dietary Patterns and Overweight/Obesity

In linear regression models with logBMI as a dependent variable, pattern 5 (alcohol, eggs, dairy) was positively related to the dependent variable in the total sample and women (Table 3). Underreporting and age were also significant determinants of logBMI. In logistic regression models with BMI used as a dichotomous variable, no association was found between the total sample and women. In men patterns 4 (meat, poultry, fish) and 6 (fast foods, total sweets) were positively related to the possibility of overweight/obesity (Table 4) The Nagelkerke R^2^ values of the model for the total sample, men and women were 16.4%, 29.3% and 15.9% respectively.

#### 3.4.2. Dietary Patterns and Waist Circumference

In linear regression models with 1/WC as the dependent variable, patterns 4 (meat, poultry, fish) and 6 (fast foods and sweets) were negatively associated with the ratio of 1/WC in men (Table 5). Due to missing values of WC in men and an inadequate number of subjects to form two categories, logistic regression was only performed on the total sample and women. There was no relation between the dietary patterns with the probability of high WC (data not shown).

### 3.5. Meals and Meal Score

The percentages of students consuming breakfast, morning snack, lunch, evening snack, dinner and bedtime snack everyday were 46.7, 14.5, 83.8, 23, 53.6 and 8.9%, respectively. The total meal score was not associated with BMI, WC or the probability of overweight/obesity, or increased WC in the multi-adjusted analysis. The total snack score was inversely related to BMI and WC in unadjusted analysis (rho = −0.138, *p* = 0.01 and rho = −0.761, *p* < 0.001, correspondingly). After multi-adjustment, snack scores were not significantly related to anthropometric values.

#### 3.5.1. Meal Patterns

Three meal patterns were identified with the PCA analysis, namely, “early eater” (consuming breakfast, morning snack and afternoon snack), “medium eater” (consuming lunch and dinner) and “late eater” (consuming bedtime snack) (Appendix A). In other words, for “early eaters”, the frequencies of breakfast, snack and afternoon snack were correlated, whereas for “medium eaters” the frequencies of lunch and dinner were correlated. The pattern of “late eaters” was formed from students who had a high consumption of bedtime snacks. It is noted that the “early eater” pattern was positively associated with MedDietScore after adjustment for age and BMI in both genders (rho = 0.301, *p* = 0.01 for men and rho = 0.219, *p* < 0.001 for women).

#### 3.5.2. Meal Patterns and Overweight/Obesity

In models of linear regression with BMI as a continuous variable, meal patterns were not significant in “predicting” the dependent variable. In models of logistic regression, men following the “early eater” pattern had a lower probability of being overweight (model’s Nagelkerke R^2^ = 27.3%) after adjusting for age, living area, department and underreporting, while in women the meal pattern was not significant (Table 6).

#### 3.5.3. Meal Patterns and Waist Circumference

Meal patterns were insignificant in “predicting” WC both in linear regression and in logistic regression models when the whole sample was considered or in a sex-specific analysis.

## 4. Discussion

In the present study, several associations of dietary and meal patterns with obesity and central obesity were documented. Patterns rich in protein, fast foods and sweets were positively related to the possibility of overweight/obesity and WC in men. In the total sample and women, subjects who tended to consume more alcohol, eggs and dairy also had a higher BMI. Regarding meal patterns, “early eaters” (men) had a lower probability of being overweight.

Males had higher obesity rates than females, which is a common finding among Greek students [30,42] or students in other countries [43,44,45,46]. The percentage of overweight male students (30%) was relatively high and similar to those reported in Greek [31], Egyptian [47] or US [48] populations, higher than those reported for Croatian [49], UK [50], Tunisian [15], Saudi Arabian [51], Lebanese [52], Syrian [53] and Spanish [13] students and lower than that reported for Australian students [54]. As far as overweight female students are concerned, their proportion (13%) was similar to that reported for Greek female students (13.7%) [55] or higher (9.4%) [18].

The median intake of fruits, vegetables, grains, dairy, poultry, fish and olive oil was lower than the National Greek recommendations, while the meat intake was higher than that recommended [56]. The median eggs and legumes intake (only for men) was in accordance with the recommendations [56]. A high meat intake has been reported in other studies in the same age group [57]. The low absolute intakes of several food groups may also be due to errors in estimating dietary intake, such as in the case of underreporting, which was high in our sample. It is, however, noted that in multi-adjusted analysis, underreporting was accounted for as a covariate.

Overall, dietary patterns better reflect the diet–health relationship than single foods [58]. The MedDietScore, representing the adherence to the Mediterranean diet, has been inversely related to risk factors, indicating the importance of having a “total diet approach” [38]. In parallel, dietary patterns identified by PCA analysis have been inversely related to inflammatory indices [59] and health conditions, such as central obesity [60]. In the present study, the adherence to the MedDietScore was higher than that documented in Greek students [18] and Greek young navy recruits (mean age 22.5 years) [61]. Compared to the general Greek population, the investigated sample had higher MedDietScore scores than those previously reported [38] (present sample means of 29.4 for men and 30.0 for women vs ATTICA study means of 25.4 for men and 27.1 for women) [38] but lower scores than elderly Greeks [62]. Studies on university students in other countries have shown a lower adherence to the Mediterranean diet [18,53]. However, direct within-study comparisons have shown a higher Mediterranean diet adherence of Dutch students than Greeks [18]. Some studies have shown no association of the Mediterranean diet to anthropometric indices [53], while others have shown inverse relationships between obesity and Mediterranean Diet adherence [63]. In our study, there was no linear relationship between the Mediterranean diet and BMI in multi-adjusted linear regression, but an inverse monotonic relationship was found. It is also noted that the MedDietScore has been positively associated with other dietary quality indices in the present population, such as the Food Compass Score, as recently described [34].

In a posteriori analysis, we found that several patterns (meats, alcohol–eggs, fast foods–sweets) were related to obesity and/or central obesity. Although the direct comparison of our results to those of other studies is rather difficult given the data-driven nature of such analysis, high-fat and high-sugar foods have previously been connected to obesity in students [46,64,65] and young adults [66]. High alcohol intake has also been associated with obesity in other studies, including university students [63] but not all [46]. The documented correlation of a high protein pattern with obesity is in line with the previously reported metabolic risks of this pattern [57]. Our results are of additive value given that a posteriori-defined unhealthy eating patterns are not usually associated with central obesity [60].

Along with food-derived clusters, another angle of the diet that deserves attention is the meal-level information [67], although there is no consensus on the definition of snacks or meals [24]. Although the majority of young Greek adults (20–50 years old) tend to eat snacks [68], in the present study, everyday snacking was present in less than 25% of students. Snack intake has been associated with >2 metabolic syndrome risk factors in Greek university students, which implies that the quality of ingested snacks is important [33]. Meal frequency has been associated with body fat in Spanish students [63]. In our study, no association was found between the meal and/or snack scores and overweight/obesity. However, from a more sophisticated analysis of meal patterns [67], we found that early eaters had reduced rates of adiposity measures. The protection of early eaters against obesity may be related to the quality of the diet as documented in our and other studies [69] or the effect of chrono-nutrition [70]. Meals consumed in the evening (by late eaters) have been connected to lower resting metabolic rates and obesity, as recently reviewed [70], while late eating has also been linked to the reduced efficacy of a weight-loss intervention, worse cardiometabolic profile and other obesogenic behaviors [71]. Late eating is conceptually connected to skipping breakfast, a habit that has also been related to obesity [72]. Mistimed eating can also result in the dysregulation of hormones involved in energy metabolism (such as leptin, cortisol, adiponectin, pro-opiomelanocortin, gastric inhibitory polypeptide and others), which can in turn promote obesity [73].

The MedDietScore was positively associated with the “early eater” phenotype, which is in line with the results from other studies showing a positive relationship with “eating breakfast” [19,74]. Indeed, several correlates of the adherence to the Mediterranean diet have been documented in university students. Low adherence to the Mediterranean diet has been associated with higher perceived stress, lower fruit and vegetables intake [17] and risky behaviors in university students, such as alcohol intake, while diet quality has been connected to a better lifestyle, moderate alcohol intake, no smoking, better sleep, etc. [63]. In this context, the beneficial feature of breakfast eating is associated with a general healthy eating profile, such as that adhering to the Mediterranean diet.

The strengths of our study include the use of a satisfactory sample of university students from a non-represented area in the literature as well as the robust analysis, by including both a priori and a posteriori ways of capturing diet in terms of foods and meals and adjusting for underreporting.

Several limitations of our study should be considered along with the interpretation of our results. First, the cross-sectional design of our study cannot reveal causal relationships between the investigated parameters. Second, the generalizability of the presented results may be limited since our sample included university students in the area of Peloponnese and data-driven statistical methods have been used. Moreover, 44% of the sample were freshmen. This means that some habits might have been “carried” to university and be connected to adiposity and not “born” at the university setting. Indeed, adiposity may be a longer-term process starting from previous years (i.e., school years). Several mistakes may have arisen in the estimation of dietary intake. However, the present study has partially accounted for dietary assessment errors since underreporting was assessed and entered in various multi-adjusted models. With respect to meal patterns, it should be noted that there is no consensus about what is a snack or meal [24], so different results may occur if different assumptions are made. Moreover, we have no data on the content/quality of meals consumed or meal intervals, which is a general caveat in studies addressing meal frequency. Finally, physical activity status was not available for the participants of the study, which could potentially influence the observed correlations.

## 5. Conclusions

In conclusion, Greek students at the University of the Peloponnese have relatively poor dietary habits, which are related to an increased probability of being overweight and high waist circumference. Reinforcing the importance of Mediterranean dietary patterns and underlining the harmful effects of fast food and sugar beverages along with meal-based counseling should be basic components of targeted health promotion programs in this population

## Figures and Tables

**Table 1 diseases-10-00064-t001:** Descriptive characteristics of participants.

	Total	Men	Women	*p*
**Age (years)**	19.61 ± 3.15	19.35 ± 1.97	19.68 ± 3.41	0.2
**Year of study**				
*1st*	153 (44.2%)	39 (50.65%)	114 (42.38%)	0.2
*2nd*	56 (16.2%)	23 (29.87%)	33 (12.27%)	<0.001
*3rd*	63 (18.2%)	10 (12.99%)	53 (19.7%)	0.2
*4th*	72 (20.8%)	4 (5.19%)	68 (25.28%)	<0.001
Living area (before enrollment)				
*<50,000 inhabitants*	79 (28.6%)	16 (36.7%)	63 (29.2%)	0.2
*>50,000 inhabitants*	197 (71.4%)	44 (73.3%)	153 (70.8%)	0.3
**Department**				
*Nursing* *, n (%)*	120 (34.7%)	22 (28.57%)	98 (36.43%)	0.2
*History, Archaeology and Cultural Resources Management, n (%)*	71 (20.5%)	19 (24.68%)	52 (19.33%)	0.3
*Philology* *, n (%)*	88 (25.4%)	17 (22.08%)	71 (26.39%)	0.4
*Sports Organization and Management, n (%)*	15 (4.3%)	3 (3.89%)	12 (4.47%)	0.8
*Other* *, n (%)*	52 (15%)	16 (20.78%)	36 (13.38%)	0.1
**ΒΜΙ (kg/m^2^) ^a^**	22.0 (19.9–24.4)	23.7 (21.4–25.9)	21.6 (19.7–23.8)	0.002
*Normal weight* *, n (%)*	238 (69%)	49 (64.5%)	189 (70.3%)	0.3
*Overweight* *, n (%)*	58 (16.8%)	23 (30%)	35 (13%)	<0.001
*Obese* *, n (%)*	10 (2.9%)	1 (1.3%)	9 (3.3%)	0.3
*Morbid obese* *, n (%)*	2 (0.6%)	0 (0%)	2 (0.7%)	0.4
**Waist circumference (cm) ^a^**	80.0 (74.0–89.0)	86.0 (79.0–90.7)	78.0 (72.0–89.0)	<0.001
*Increased waist circumference* *, n (%)*	108 (37.8%)	13 (19.1%)	95 (43.6%)	<0.001

**^a^** The variable was transformed prior to statistical comparisons to achieve normality. Data are presented as mean ± standard deviation for normally distributed variables. Otherwise, data are presented as median (lower–upper quartile) (25th–75th). Categorical variables are displayed as n (valid %). Student *t*-test, Mann–Whitney or Chi-square test (for categorical variables) were used to compare means.

**Table 2 diseases-10-00064-t002:** Dietary intake of the participants.

Servings per Day or Week	Total	Men	Women	*p*
**Full-fat dairy/day †**	0.50 (0.10–1.12)	0.66 (0.18–1.39)	0.45 (0.08–1.05)	0.02
**Low-fat dairy/day †**	0.75 (0.24–1.45)	0.61 (0.24–1.30)	0.79 (0.24–1.51)	0.3
**Total dairy/day †**	1.46 (0.90–2.21)	1.41 (0.86–2.34)	1.46 (0.91–2.15)	0.8
**Fruits (1 item)/day**	0.96 (0.42–1.91)	0.83 (0.31–1.47)	0.99 (0.45–2.03)	0.07
**Vegetables (1 cup raw, ½ cup boiled)/day**	2.44 (1.41–3.94)	1.74 (1.06–3.50)	2.51 (1.59–4.05)	0.01
**Legumes (1 cup)/week**	2.31 (1.14–3.81)	1.58 (0.74–4.15)	2.31 (1.28–3.75)	0.1
**Fish and sea foods (170 g)/week**	0.51 (1.04–2.03)	1.02 (0.46–2.13)	1.07 (0.58–1.99)	0.6
**Red meat (125 g)/week**	6.90 (4.10–11.30)	9.75 (6.19–15.78)	6.18 (3.92–10.23)	<0.0001
**Poultry (125 g)/week**	1.16 (0.58–1.99)	1.16 (0.58–2.11)	1.16 (0.58–1.99)	0.6
**Whole-wheat grains and products/day ‡**	0.08 (0.31–0.86)	0.20 (0.07–0.88)	0.12 (0.02–0.51)	0.2
**Refined grains and products/day ‡**	1.22 (2.03–3.08)	2.33 (1.44–3.51)	1.97 (1.15–2.99)	0.05
**Eggs (1 item)/week**	0.58 (1.16–2.19)	1.98 (1.07–3.86)	1.16 (0.46–1.99)	0.001
**Nuts (40 g)/day**	0.02 (0.04–0.16)	0.11 (0.02–0.28)	0.03 (0.02–0.11)	<0.001
**Sweets (1 piece of cake, 5 biscuits)/day**	1.7865 (0.9–3.13)	1.83 (0.69–3.44)	1.69 (0.99–3.02)	0.8
**Honey and marmalades** **/day**	0.16 (0.03–0.57)	0.12 (0.02–0.51)	0.22 (0.04–0.59)	0.02
**Olive oil/day**	1 (0.5–1)	0.78 (0.50–1.0)	1.0 (0.50–1.0)	0.6
**Other fats/day**	0.24 (0.06–0.66)	1.04 (0.79–1.71)	1.08 (0.86–1.5)	0.3
**Fast foods/day**	0.08 (0.03–0.16)	0.09 (0.04–0.29)	0.06 (0.02–0.16)	0.06
**Juices/day**	0.60 (0.26–1.08)	0.54 (0.25–0.92)	0.65 (0.27–1.10)	0.1
**Sodas/day**	0.18 (0.05–0.51)	0.33 (0.06–0.68)	0.17 (0.04–0.44)	0.02
**Liquid calories/day**	0.90 (0.46–1.45)	0.93 (0.50–1.42)	0.86 (0.45–1.45)	0.6
**Alcoholic drinks** **/day ***	0.19 (0.04–0.56)	0.30 (0.09–0.68)	0.17 (0.04–0.51)	0.02
**Energy/BMR**	1.56 (1.09–2.19)	0.99 (0.72–1.42)	1.76 (1.3–2.48)	<0.001
**Underreporting (%)**	24.9	54.5	16.4	<0.001

Data are presented as median (lower–upper quartile) (25th–75th) as parameters were not normal. Categorical variables are displayed as n (%). Mann–Whitney test and Chi-square test (for categorical variables) were used to compare means. † One serving of dairy was considered 250 mL milk, 200 g yogurt, 30 g cheese; ‡ one serving of grains and products was considered 1 slice of bread, 2 melba toasts, ½ cup pasta; ***** one serving of alcoholic drinks was considered 125 mL wine, 330 mL beer or 40 mL whiskey.

**Table 3 diseases-10-00064-t003:** Linear regression analysis for logBMI in the total sample, men, and women.

	Total Sample adj R^2^ = 15.8%			Men adj R^2^ = 8%			Women adj R^2^ = 15.5%		
	B	SE	*p*	B	SE	*p*	B	SE	*p*
(Constant)	1.283	0.031	<0.001	1.312	0.078	<0.001	1.233	0.029	<0.001
***Gender (men* vs. *women *)***	−0.014	0.010	0.1	NA	NA	NA	NA	NA	NA
** *Age (years)* **	0.003	0.001	0.005	0.002	0.004	0.5	0.003	0.001	0.01
** *Department (nursing vs. others)* **	0.005	0.003	0.09	0.012	0.006	0.06	0.002	0.003	0.4
** *Living area (before enrollment) †* **	0.002	0.008	0.8	−0.019	0.019	0.3	0.009	0.009	0.3
** *Pattern 1* ** *: legumes, vegetables, fruits, tea, dairy, whole grains*	0.005	0.004	0.1	0.012	0.008	0.1	0.002	0.004	0.6
** *Pattern 2* ** *: juices, sodas, liquid calories*	0.000	0.004	0.9	−0.008	0.009	0.3	0.001	0.004	0.7
** *Pattern 3* ** *: olive oil, fats*	−0.004	0.004	0.2	0.007	0.010	0.4	−0.007	0.004	0.08
** *Pattern 4* ** *: meat, poultry, fish*	−0.002	0.004	0.6	0.000	0.006	0.9	0.000	0.007	0.9
** *Pattern 5* ** *: alcohol, eggs, dairy*	**0.008**	**0.003**	**0.02**	0.004	0.010	0.7	**0.009**	**0.004**	**0.02**
** *Pattern 6* ** *: fast foods, total sweets*	−0.004	0.004	0.2	0.000	0.009	0.9	−0.006	0.004	0.1
** *Underreporting (yes */no)* **	0.044	0.011	<0.001	0.037	0.026	0.1	0.052	0.013	<0.001

† 1 = <50,000 inhabitants, 2 = >50,000 inhabitants; the dietary parameters related to the dependent variable are shown in bold. B: unstandardized beta coefficient; SE: standard error; NA: not applicable; * reference category.

**Table 4 diseases-10-00064-t004:** Logistic regression analysis for overweight/obesity in the total sample, men and women.

	Total Sample					Men					Women				
			95% CI					95% CI					95% CI		
	B	Exp(B)	Lower	Upper	*p*	B	Exp(B)	Lower	Upper	*p*	B	Exp(B)	Lower	Upper	*p*
(Constant)	−0.239	0.788			0.8	3.291	26.8			0.2	−1.489	0.226			0.2
** *Gender (men vs. women *)* **	0.163	1.177	0.525	2.641	0.6	NA	NA	NA	NA	NA	NA	NA	NA	NA	NA
** *Age (years)* **	0.030	1.030	0.950	1.117	0.4	−0.025	0.975	0.730	1.302	0.8	0.036	1.037	0.948	1.133	0.4
** *Department (nursing vs. others *)* **	−0.314	0.731	0.350	1.524	0.4	−0.611	0.543	0.107	2.762	0.4	−0.278	0.757	0.309	1.854	0.5
** *Living area (before enrollment) †* **	−0.250	0.779	0.372	1.629	0.5	−1.403	0.246	0.053	1.131	0.07	0.278	1.321	0.505	3.454	0.5
** *Pattern 1* ** *: legumes, vegetables, fruits, tea, dairy, whole grains*	0.200	1.222	0.868	1.720	0.2	0.459	1.583	0.808	3.099	0.1	−0.007	0.993	0.605	1.629	0.9
** *Pattern 2* ** *: juices, sodas, liquid calories*	0.161	1.175	0.851	1.621	0.3	−0.124	0.883	0.429	1.818	0.7	0.262	1.300	0.891	1.896	0.1
** *Pattern 3* ** *: olive oil, fats*	0.251	1.285	0.917	1.800	0.1	0.778	2.177	0.964	4.915	0.06	0.119	1.126	0.735	1.726	0.5
** *Pattern 4* ** *: meat, poultry, fish*	0.087	1.091	0.799	1.489	0.5	**0.463**	**1.589**	**0.995**	**2.538**	**0.05**	−0.145	0.865	0.419	1.784	0.6
** *Pattern 5* ** *: alcohol, eggs, dairy*	0.242	1.273	0.944	1.718	0.1	0.191	1.211	0.446	3.285	0.7	0.371	1.450	0.948	2.217	0.08
** *Pattern 6* ** *: fast foods, total sweets*	0.234	1.264	0.903	1.769	0.1	**0.937**	**2.552**	**1.074**	**6.060**	**0.03**	−0.026	0.974	0.599	1.585	0.9
** *Underreporting (yes */no)* **	−1.813	0.163	0.064	0.415	<0.001	−3.181	0.042	0.003	0.519	0.01	−1.664	0.189	0.064	0.562	0.003

CI: confidence interval; NA: not applicable; * reference category; † 1 = <50,000 inhabitants, 2 = >50,000 inhabitants (reference category); the dietary parameters related to the dependent variable are shown in bold. Exp(B): exponentiation of the B coefficient (odds ratio).

**Table 5 diseases-10-00064-t005:** Linear regression analysis for l/WC in the total sample, men and women.

	**Total Sample** **adj R^2^ = 15.9%**			**Men** **adj R^2^ = 28.7%**			**Women** **Adj R^2^ = 10.1%**		
	**B**	**SE**	** *p* **	**B**	**SE**	** *p* **	**B**	**SE**	** *p* **
(Constant)	0.013	0.001	<0.001	0.013	0.002	<0.001	0.015	0.001	<0.001
***Gender (men* vs. *women *)***	0.001	0.00003	0.03	NA	NA	NA	NA	NA	NA
** *Age (years)* **	−5.0 × 10^−5^	0.00003	0.1	−5.3 × 10^−5^	9 × 10^−5^	0.5	−4.75 × 10^−5^	4 × 10^−5^	0.2
** *Department (nursing vs. others)* **	0.0003	0.00009	0.002	−0.0004	0.0001	0.02	−0.0002	0.0001	0.03
** *Living area (before enrollment) †* **	5.8 × 10^−5^	0.0001	0.8	0.001	0.0004	0.06	−0.0001	0.0003	0.6
** *Pattern 1* ** *: legumes, vegetables, fruits, tea, dairy, whole grains*	−3.11 × 10^−5^	0.0001	0.8	−0002	0.00031	0.5	−4.05 × 10^−5^	0.0001	0.8
** *Pattern 2* ** *: juices, sodas, liquid calories*	−4.4 × 10^−5^	0.0001	0.7	−0.003	0.0002	0.2	−4.07 × 10^−5^	0.0001	0.7
** *Pattern 3* ** *: olive oil, fats*	−3.7 10^−6^	0.0001	0.9	−0.0004	0.0002	0.08	0.00011	0.0001	0.5
** *Pattern 4* ** *: meat, poultry, fish*	1.8 × 10^−4^	0.0001	0.1	**−0.000** **3**	**0.000** **1**	**0.02**	−0.000127	0.00026	0.6
** *Pattern 5* ** *: alcohol, eggs, dairy*	1.6 × 10^−4^	0.0001	0.2	−8.5 × 10^−5^	0.0002	0.7	0.000174	0.00019	0.3
** *Pattern 6* ** *: fast foods, total sweets*	7.1 × 10^−5^	0.0001	0.5	**−0.001**	**0.000** **2**	**0.01**	0.000241	0.000154	0.1
** *Underreporting (yes */no)* **	−0.001	0.0001	0.005	−0.002	0.001	0.01	−0.001	0.000	0.004

† 1 = <50,000 inhabitants, 2 = >50,000 inhabitants; the dietary parameters significantly related to the dependent variable are shown in bold. B: unstandardized beta coefficient; SE: standard error; NA: not applicable; * reference category.

**Table 6 diseases-10-00064-t006:** Logistic regression analysis with overweight/obesity as the dependent variable and meal patterns as the independent variable in the total sample, men and women.

	Total Sample					Men					Women				
		Exp(B)	95% CI					95% CI					95% CI		
	B		Lower	Upper	*p*	B	Exp(B)	Lower	Upper	*p*	B	Exp(B)	Lower	Upper	*p*
(Constant)	−0.016	0.984			0.9	1.460	4.307			0.6	−1.514	0.220			0.3
** *Gender (men vs. women *)* **	−0.588	0.555	0.249	1.238	0.1	NA	NA	NA	NA	NA	NA	NA	NA	NA	NA
** *Age (years)* **	0.013	1.013	0.913	1.125	0.8	−0.071	0.932	0.699	1.242	0.6	0.016	1.016	0.907	1.139	0.7
** *Department (nursing vs. others *)* **	−0.258	0.772	0.368	1.621	0.4	0.449	1.567	0.983	2.497	0.05	0.141	1.151	0.858	1.544	0.3
** *Living area (before enrollment) †* **	0.229	1.258	0.992	1.594	0.05	−1.030	0.357	0.084	1.517	0.1	0.181	1.198	0.462	3.104	0.7
** *Early eaters* **	−0.336	0.715	0.503	1.015	0.06	−0.747	0.474	0.220	1.020	**0.05**	−0.187	0.830	0.542	1.270	0.3
** *Medium eaters* **	−0.008	0.992	0.730	1.349	0.9	0.251	10.286	0.591	2.795	0.5	−0.087	0.917	0.663	1.267	0.5
** *Late eaters* **	0.105	1.110	0.796	1.549	0.5	0.120	1.127	0.605	2.102	0.7	0.173	1.189	0.769	1.840	0.4
** *Underreporting (yes */no)* **	−1.031	0.357	0.168	0.756	0.007	−0.347	0.707	0.172	2.910	0.6	−1.418	0.242	0.097	0.607	0.002

CI: confidence interval; Exp(B): exponentiation of the B coefficient (odds ratio); NA: not applicable; * reference category; † 1 = <50,000 inhabitants, 2 = >50,000 inhabitants (reference category).

## Data Availability

Data are available upon request.

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
