# Peer review of "Mediterranean Diet, a Posteriori Dietary Patterns, Time-Related Meal Patterns and Adiposity: Results from a Cross-Sectional Study in University Students"

_diseases, 2022, doi:10.3390/diseases10030064_

Round 1
Reviewer 1 Report
Congrats to the authors for completing the work. Here are my review comments to improve the draft further -
* Why are alcohol and eggs combined together in Factor 5? Any rationale? Does this mean that these participants consumed only alcohol and eggs?
* Recommend rewriting the entire draft for better readability, especially the Methodology and Results section needs to be condensed, and the writing made crisper.
I have also highlighted sections that are especially unclear in the pdf, with comments
* Factor 1,2,3,4,etc or Pattern 1,2,3,4,etc? - Suggest using one form uniformly throughout the draft, in text and in tables
* Table 1 - "missing" row can be removed from the table. The information can be mentioned as N in parenthesis next to the total, or in the text.
* Table 2 - The first column seems too wordy
* Table 3 - Factor 1 or Pattern 1?
* Tables - Capitalize NA
* Table 5 - Expand B and SE in the legend
* Meal patterns: Line 230 - Did the early eaters not have dinner or bedtime snacks? This is unclear and this section needs improvement
* Discussion: Discuss the relation between early or late-eaters and being overweight/obesity
* Section 3.5.3 - rephrase or skip
* Lines 261-264: check comments in pdf
* Lines 267-268: does it refer to female students? please specify
* Line 318: How so? Please mention
Best wishes,

Reviewer 2 Report
This work examines dietary habits of Greek college students and parameters of adiposity on the basis that the new environment presents possible challenges to any diet habits or reinforces bad habits. Some features of the study are a little confusing. Why was the work performed in 2013, if I read the paper correctly (line 63), and not reported until now? Second, based on the dates of data collection, some of the 1st year students would have just started the term, so diet practice at the school would not have had a chance to have much effect. Diet habits established prior to term would be the cause of any measured adiposity factors. This is not a fatal flaw, it just requires some adjustment in the background for the study, indicating that some habits might have carried to school and not a result of poor choices made in school. The school just provided a convenient local to study diet habits in a generally young population. To make the point, you are suggesting in the introduction, the 2nd year and higher students would be the better choice for analysis.
A second issue is the close wording between the 2022 paper on the Food compass score. Some more editing should be done to lose the resemblance of the two papers.
Round 2
Reviewer 1 Report
The review comments have been satisfactorily addressed and the manuscript looks largely improved.
Best wishes
Reviewer 2 Report
The revised paper is acceptable for publication.